# Interactive Simulation of Nonpharmaceutical Interventions of Airborne Disease Transmission in Office Settings

**DOI:** 10.3390/ijerph21111413

**Published:** 2024-10-25

**Authors:** Thomas Zimmerman, Neha Sharma, Hakan Bulu, Vanessa Burrowes, David Beymer, Vandana Mukherjee

**Affiliations:** 1IBM Research-Almaden, San Jose, CA 95120, USA; tzim@us.ibm.com (T.Z.); hbulu@us.ibm.com (H.B.); beymer@us.ibm.com (D.B.); 2School of Computer Science, Simon Fraser University, 8888 University Dr W, Burnaby, BC V5A 1S6, Canada; neha_sharma_3@sfu.ca; 3IBM Research Triangle Park, Research Triangle Park, NC 27709, USA; vanessa.burrowes@ibm.com

**Keywords:** COVID, modeling, Wells-Riley, CO_2_, aerosol, airborne, infection, transmission

## Abstract

The COVID-19 pandemic has caused major disruptions to workplace safety and productivity. A browser-based interactive disease transmission simulation was developed to enable managers and individuals (agents) to optimize safe office work activities during pandemic conditions. The application provides a user interface to evaluate the impact of non-pharmaceutical interventions (NPIs) policies on airborne disease exposure based on agents’ meeting patterns and room properties. Exposure is empirically calibrated using CO_2_ as a proxy for viral aerosol dispersion. For the building studied, the major findings are that the cubicles during low occupancy produce unexpectedly high exposure, upgrading meetings to larger rooms reduces total average exposure by 44%, and when all meetings are conducted in large rooms, a 79% exposure reduction is realized.

## 1. Introduction

The COVID-19 pandemic has caused major disruptions to workplace conduct and activities. The most drastic containment measure has been the suspension of all office work. As the understanding of airborne disease transmission mechanisms has evolved, non-medical intervention (NPI) strategies have emerged to help people return to work while maintaining safety. General guidelines such as social distancing, mask-wearing, and enhanced ventilation are among the strategies to reduce disease transmission. However, the specifics of how NPI methods are implemented, encouraged, or mandated are often left to the discretion of company and building owners, site operations managers, executives, and individuals. Decisions on workplace transmission control strategies may not always be scientifically driven.

While sophisticated computer simulations have been developed to model the impact of NPIs, they typically require advanced technical knowledge to collect site-specific conditions, set up, tune, run, and interpret, often exceeding the skills and abilities of decision-makers (“users”) and building occupants (“agents”). Furthermore, these complex models demand significant processing power and long run times, making it difficult to test the efficacy of different NPI strategies efficiently [1]. To address these challenges, a browser-based interactive tool was designed to explore and evaluate the impact of NPI policies on agents’ exposure to airborne disease. A customizable meeting-based exposure model supported by empirical data efficiently calculates relative viral load, providing a sub-second response time. 

The approach was inspired by the work of Bazant and Bush [2], as well as McCarthy et al. [3], who developed models of airborne disease transmission deployed in a spreadsheet to provide safe exposure time recommendations for building occupants. Exposure is calculated based on user input parameters, including room volume, air exchange rates, breathing rates, and mask usage. The model assumes airborne pathogens are uniformly distributed throughout the space, following the Wells-Riley model [4]. However, fluid dynamic analysis and empirical measurements have shown that transmission distance plays a significant role in contagion risk [5]. Viral particles in fine aerosols can travel several meters and persist in the air for extended periods [6]. Generalizing the impact of transmission distance in office environments is challenging due to the significant influence of site-specific factors, such as room geometry and ventilation systems.

Recognizing the challenges of predicting the absolute risk of disease transmission due to numerous unknown variables, McCarthy et al. [3] estimated the relative risk of an activity by aggregating the risks associated with each individual sub-activity. Using this approach, the authors calculated the relative risk for common activities such as airplane seating, stadium sporting events, classrooms, and restaurants. Similar to Bazant and Bush, their model includes a customizable spreadsheet that allows users to adjust NPI parameters such as social distancing, mask wearing, room size, and ventilation to mitigate risk. However, unlike Bazant and Bush, McCarthy et al. incorporate several models that account for the decay of infection risk with distance [7,8].

In contrast to these authors, who rely on theoretical models of transmission, in this paper we collect and use site-specific empirical data to predict distance- and room-specific airborne transmission risks using CO_2_ as a surrogate for aerosolized virus [9,10,11,12,13]. 

### 1.1. CO_2_ as a Surrogate for Aerosolized Virus

Normal breathing and speech predominantly produce small droplets that are susceptible to aerosol transport. Aerosolized SARS-CoV-2 particles can remain suspended in the air for hours and can travel over distances, including outside of rooms and within buildings [14,15]. A significant proportion of pathogens are found in small particles (<5 μm) [16]. Particles with diameters of 1–3 μm can remain suspended almost indefinitely [17]. 

Air ventilation and distribution play a crucial role in reducing indoor transmission of virus-laden aerosol particles. Airflow in an enclosed room with forced ventilation creates a complex fluid dynamic system, where factors such as air supply and return vent placement, room geometry, and transient events (e.g., occupant movement and door opening) interact, influencing the distribution and movement of air and airborne particles [18,19]. 

The use of tracer gas is a common method for empirically simulating the spread of airborne diseases, although its accuracy is often debated. Ai et al. [20] highlight four key concerns: particle dynamics differ from gas due to factors such as gravity, inertia, and surface deposition; airflow with particles represents a two-phase flow (air and liquid); the coagulation and resuspension of particles cannot be replicated by gas; and particles have varying aerodynamic diameters. Nevertheless, they argue that tracer gas remains a suitable surrogate for studying fine droplet nuclei (less than 3–5 μm), which are the primary carriers of airborne pathogens, owing to its simplicity and the ease of obtaining reliable results.

In their 2017 study, Bivolarova et al. [21] used nitrous oxide (N_2_O) as a tracer gas alongside particles of three different sizes (0.07, 0.7, and 3.5 μm) to simulate aerosol distribution in a room with a thermal manikin. The tracer gas reliably predicted exposure to all particle sizes within the breathing zone, and changes in ventilation rate or room surface area did not affect the consistency in dispersion between the tracer gas and the particles. Similarly, Cui et al. [22] validated the use of CO_2_ tracer gas methods for measuring air change rates in ventilated rooms, demonstrating that CO_2_ concentrations can provide valuable insights into ventilation efficiency and airflow distribution in indoor spaces. An experimental study in a concert hall emitted an air stream containing a mixture of CO_2_ gas and an aerosol with a peak particle diameter of 0.3 um. Simultaneous measurements of CO_2_ and aerosol concentration at different locations resulted in a Pearson correlation coefficient of 0.77 [9]. 

### 1.2. Masks

Surgical masks are primarily designed to reduce the exhalation of bacteria from the mouth and nose rather than to protect against the inhalation of airborne particles. Consequently, real-world studies often report lower filtration efficiency for surgical masks. Issues such as glasses fogging demonstrate the prevalence of leaks during exhalation, particularly with pleated masks, which tend to fit less snugly compared to cup-shaped masks [23,24]. In contrast, N95 masks are rated to have a minimum efficiency of 95%, as determined by standardized tests involving the inhalation of 300 nm diameter uncharged sodium chloride aerosol particles [25,26].

### 1.3. Social Impacts of NMI

Several studies highlight the social and psychological impacts of COVID-19 safety measures in the workplace. Jahantab et al. [27] found that social distancing can lead to emotional exhaustion, negatively affecting employee performance, though strong team support mitigated this impact. Cartaud et al. [28] noted that mask-wearing may create a false sense of security, leading to reduced social distancing, which could increase transmission, especially due to leakage from poorly fitting masks. Wagemann et al. [29] explored how masks impair nonverbal communication, disrupting meaningful connections and reducing engagement in workplace interactions. Together, these studies emphasize that while social distancing and mask-wearing are critical for health protection, they can impact team dynamics, communication, and overall productivity.

In this paper, the main contributions are as follows: First, the development of an interactive browser-based application for rapidly composing and comparing NMI policies using predicted exposure. Second, a simple method to customize and calibrate the application for site-specific rooms in a target building using empirical tracer-gas measurements. Third, a method to summarize the activities of office occupants (agents) based on regularly scheduled meetings in site-specific, empirically calibrated rooms. Finally, comparing policies to a familiar reference point, such as a “Business as Usual” scenario where no NMI actions are taken, offers users a relatable way to understand the impact of NMI strategies on the predicted exposure to aerosolized infectious agents.

## 2. Materials and Methods

The interactive simulation is composed of three components. The exposure simulation implements a user interface enabling the user to manipulate NMI settings and run exposure simulations on the client’s browser. The Meeting Profile, customized by the user, describes meeting types and conditions. The Room Profile contains site-specific empirically determined room-specific data that enable the browser to calculate exposure. 

Exposure is calculated by a distant dependent mass transport model, determined by measuring the percent of CO_2_ mass released that a detector receives. Masks attenuate mass transmission by the product of their ability to block transmission and reception of virus particles. 

### 2.1. Exposure Simulation

The exposure simulation provides a graphical user interface (GUI) that allows users to create an intervention policy (Figure 1, blue values) by adjusting five NPI parameters and viewing the resulting exposure predictions in real time (<1 s response time). Behind the GUI, an exposure calculation sums the predicted exposure for an average agent attending all meetings with the specified NPI. The resulting average agent exposure is displayed as a relative value, compared to a “Business As Usual” (BAU) policy, which represents typical office conditions before the COVID-19 pandemic. 

Referring to Figure 1, the five NPI parameters are: social distancing (in meters), mask-wearing fraction (percent of agents wearing masks, 0 to 100%), room selection, meeting interval (days between meetings), and duration (in hours). The “Business As Usual” (BAU) policy is defined as a social distance of one meter, no mask-wearing, and the interval, duration, and room assignments shown in red values in Figure 1.

The browser-based application is designed as a single-page application (SPA), allowing for dynamic interaction without requiring webpage reloading. It was developed using React [30], a declarative, efficient, and flexible JavaScript library for building user interfaces. React’s component-based architecture facilitated a modular design, promoting code reusability and maintainability.

JavaScript forms the backbone of the application’s implementation, handling both the simulation algorithm’s calculations and the application logic. To ensure a responsive user experience, the algorithm’s calculations are executed in real-time on the client side. This approach minimizes latency by eliminating the need for server-side computation and data retrieval, providing immediate feedback to the user as input parameters are adjusted.

The application’s user interface was designed to be intuitive and user-friendly, catering to both technical and non-technical audiences. Input parameters for the simulation are adjusted using sliders, enhancing user engagement and making it easy to explore different scenarios without requiring direct numerical input. The immediate visual feedback provided when adjusting parameters encourages interactive exploration of the model’s behavior under various conditions.

To effectively communicate the outcomes of the algorithm’s calculations, React Google Charts [31] are integrated into the application, enabling dynamic data rendering in an easily interpretable visual format. The charts update in real-time as the simulation’s input parameters are modified, allowing users to visually analyze the effects of different parameter settings on airborne disease exposure.

The application’s performance was a paramount consideration throughout the development process. By executing the simulation algorithm entirely in JavaScript and leveraging the efficient Document Object Model (DOM) manipulation capabilities of React, we achieved significant computational efficiency. The application can process and reflect changes in the simulation parameters in real-time, with updated charts rendered in less than a second. This efficiency ensures a seamless user experience, even on devices with limited computing resources.

The simulation relies on two site-specific files, the Meeting Profile and Room Profile, to perform the exposure predictions. The Meeting Profile is created and customized by the user, based on the type of meetings agents typically attend. The Room Profile contains site-specific data, collected from tracer gas experiments in the actual rooms where meetings occur. 

### 2.2. Meeting Profile

The Meeting Profile was customized for nine types of common meetings occurring in any of six representative rooms in the research building, located in San Jose, CA, USA. The Meeting Profile (Table 1, left) defines nine types of meetings commonly occurring for agents in the building. Each meeting is characterized by its Interval (days between meetings, e.g., 0.2 means five times per day), Duration (hours), and Room Model. 

### 2.3. Room Profile

The Room Profile includes data from tracer gas measurements that enable the calculation of predicted aerosol exposure based on inter-agent distance (Social Distance). For each room, second-degree polynomial coefficients represent the mass transfer fraction, indicating the fraction of CO_2_ released that reaches an agent as a function of distance (Table 1, right). Distances greater than MaxD are set to MaxD, and a mixed-model is assumed.

### 2.4. Aerosol Exposure Calculation

A mass transfer model was developed to calculate the transfer of CO_2_ mass m(i,j) received by a susceptible agent j from an infected agent i, attenuated by physical factors that reduce transmission:m(i,j) = tx(i) × maskTx(i) × rcv(j) × maskRcv(j) × Md(i,j) × t(1)
where:tx(i) is the infectious status of agent i (1 if infectious, 0 if not infectious),maskTx(i) is the mask attenuation factor for agent i (1 if no mask, 0.05 if wearing a mask [32]),rcv(j) is the susceptibility of agent j (1 if susceptible, 0 if not susceptible),maskRcv(j) is the mask attenuation factor for agent j (1 if no mask, 0.15 if wearing a mask [32]),Md(i,j) is the mass transfer fraction (in units of ppm/gram) from agent i to agent j, determined empirically by releasing CO_2_ in rooms, andt is the amount of time (in hours) that agents i and j are co-located in the same

The simulation calculates the mass transfer Equation (1) for nine types of meetings that can occur in six types of rooms. The duration (in hours) and interval of occurrence (in days between meetings) can be adjusted through the GUI interface. The average agent exposure is calculated by summing the exposure for all meeting types, adjusted by the percentage of agents wearing masks. This process is illustrated by the following pseudo-code:
**#** Define constantsMASK_TX = 0.05 (95% effective) [32]MASK_RCV = 0.15 (85% effective) [32]
**#** Define input variablesd = Social_Distance (in meters)Mask = 0 to 1 where 0 = no mask wearing, 1 = 100% wearing, for Lunch meeting Mask = 0A, B, C = mass transfer fraction polynomial coefficients for room (from CO_2_ room measurements)Duration = meeting length of time (hours) Interval = days between meetings (e.g., 0.2 = 5 times per day)
**#** Calculate distance factordistanceFactor = A × d^2 + B × d + C
**#** Calculate exposure timeexposureTime = Duration/Interval
**#** Calculate mask factorif Mask == 1:       # masks are being worn  maskFactor = MASK_TX × MASK_RCV × Mask + (1 − Mask)else:  maskFactor = 1    # no masks are being worn (e.g., during lunch)
**#** Calculate average exposureroom_exposure = maskFactor × distanceFactor × exposureTimeNote that during lunch, all masks are forced off (Mask = 0).

### 2.5. Tracer Gas Measurements

The mass transfer fraction depends on room geometry, airflow characteristics, and the distance between infectious and susceptible individuals. These factors were directly measured using a CO_2_ gas generator as a surrogate for aerosol viral particles released by an infectious agent, along with four data-logging CO_2_ sensors simulating viral inhalation by susceptible individuals. CO_2_ dispersion measurements were conducted in six spaces: three meeting rooms of various sizes, a cafeteria, an auditorium, and a room filled with cubicles. 

Room and air circulation specifications are provided in Table 2. The rooms operate with variable airflow depending on cooling needs. Airflow and air exchanges per hour (ACH) decrease when the rooms are unoccupied, as fewer occupants generate less heat, reducing cooling demand. Tracer gas measurements were conducted with no humans present in the rooms, so the ACH was at their nominal minimum settings; 25% for all rooms except the large room, which is set at 50% due to its proximity to a large kitchen.

In each space, 400 to 500 g of dry ice was sublimated (converted from solid to gas) within 10 min, producing 0.207 to 0.259 m^3^ of gas. CO_2_ concentration (in parts per million) was sampled at one-minute intervals from four locations within the room for 30 min using CO_2_ sensors. Four sets of measurements were conducted for the auditorium, cafeteria, medium, and large rooms, three sets for the small room, and two for the cubicles. 

#### 2.5.1. CO_2_ Gas Generator

The CO_2_ generator (Figure 2) produces a plume of rising air and CO_2_ mixture (Figure 3), which is dispersed throughout the room by air currents. The 2-L Pyrex beaker of the generator is filled with crushed dry ice. The mass is wirelessly monitored with a scale (Arboleaf Model CK10G) with insulation to protect it against the extreme cold of dry ice (−78 °C). A 1500 W hair dryer positioned above the beaker directs a stream of hot air onto the dry ice, sublimating and heating the released CO_2_ gas. Without this warming, the dense cold CO_2_ gas would sink to the ground [33]. 

#### 2.5.2. CO_2_ Meter Placement

CO_2_ concentration at four locations in each room was monitored using battery-powered data-logging CO_2_ sensors (Model M2000C-2nd Generation, Elitech). Each sensor samples CO_2_ concentration (in ppm) once per minute, timestamps each reading, and stores the data in its local memory. After the experiments, the data sets from each sensor are downloaded via USB as CSV files to a laptop for analysis. 

In the small, medium, and large rooms, the CO_2_ generator was placed at the front of the room (Figure 4A–C, indicated with yellow ovals), where a speaker would typically stand, on a cart, with the top of the beaker 138 cm above the ground, and the four CO_2_ sensors were evenly spaced on the table (Figure 4A–C, indicted with green rectangles). In cubicle room the CO_2_ generator was place on the table of a corner cubical (Figure 5, indicated with yellow squares) with 4 CO_2_ sensors place on desks in adjacent cubicles lining the walls (Figure 5, indicated with blue squares, with numbers identifying each sensor). Each cubicle measures 210 × 230 cm. In the auditorium (Figure 4D), the CO_2_ generator was placed at the front of the hall, with the top of the beaker at eye level to the first row, with the four CO_2_ sensors placed in successive rows, aligned with the generator. In the cafeteria (Figure 4E), the generator was placed on a round table, and the CO_2_ sensors were positioned on adjacent tables at radial distances of 2.3 and 3.25 m, at 0 and 90 degrees, to detect any airflow asymmetry.

#### 2.5.3. Tracer Gas Protocol

For each room, the following measurement protocol was followed. The CO_2_ generator (Figure 5, yellow box) and the four CO_2_ sensors (Figure 4, green boxes) were placed in their designated positions and turned on. Crushed dry ice was poured in a glass beaker mounted on a wireless digital scale. The experimenter recorded its starting mass, left the room, and remotely plugged in the hair dryer and monitored the mass of the dry ice over Bluetooth with a phone app provided by the scale vendor. When the scale reading indicated that the dry ice had completely sublimated (approximately 10 min), the hair dryer was remotely unplugged. The CO_2_ plume dispersed primarily through air currents, as CO_2_ diffusion in air is extremely slow, estimated at 1.62 × 10^−5^ m^2^/s [9]. 

After 30 min, the experimenter re-entered the room and repeated the experiment. Once all measurements were completed, the CO_2_ concentration data (in ppm) was uploaded from the CO_2_ sensors to a laptop as CSV files via a USB interface. For each experiment run, the CO_2_ concentration values were zeroed by subtracting the initial CO_2_ concentration at the beginning of the run. The resulting values were then summed over the 30-min experiment duration and divided by the initial dry ice mass to account for small differences in starting CO_2_ mass, yielding a normalized fraction of CO_2_ received at each sensor (ppm/gram). Finally, the normalized fractions from multiple runs were averaged.

#### 2.5.4. Mass Transfer Fraction Extrapolation

For each room, the average mass transfer fraction was fitted with a second-order polynomial to generate coefficients to model the rooms. To provide mass transfer fraction values down to one meter, the polynomial curves were extrapolated using the mass percentage probability for aerosol droplets derived by Sun and Zhai [6].
Pd = (−18.19 × ln(d) + 43.276 )/100(2)

For distances beyond the location of the most distant sensor, the mass percentage value for the most distant sensor was applied, assuming a uniform distribution of aerosol viral particles was assumed (i.e., Wells-Riley mixed-model). All measurements were conducted in unoccupied rooms to prevent confounding the CO_2_ measurements with human respiration and to avoid exposing individuals to high levels of CO_2_.

## 3. Results

### 3.1. Room Calibration

The mass transfer fraction of CO_2_ (ppm/gram) detected as a function of distance was determined for six rooms (Figure 6) calculated from the data collected from tracer gas experiments (Figure 7). 

### 3.2. NPI Parameter Exploration

To examine the impact of room assignments on relative exposure, the room allocations were systematically changed for five new policies (Table 3) while keeping social distance set at 2 m, which is the distance between cubicles. The policies and the results are as follows:Policy 1: BAU reference policy, with Social Distance = 1.Policy 2: BAU room assignments were used, with Social Distance = 2, resulting in 82% exposure compared to BAU.Policies 3 to 5: Upgrade Duo, Team, and Group meetings individually to the next larger room, except for Solo meetings, where the room does not impact exposure.Policy 6: All meetings were conducted in the large room, resulting in 19% exposure compared to BAU (a 79% reduction in average agent exposure).

To compare the impact of social distancing and masking combinations, seven social distances (0.5, 1, 1.5, 2, 2.5, 3, and 4 m) were evaluated over the full range of mask wearing (0 to 100%) (Figure 8). Without wearing masks, social distancing of 2 and 3 m reduces average agent exposure by 24% and 38%, respectively, demonstrating the effectiveness of social distancing in reducing exposure.

## 4. Discussion

### 4.1. Analyzing Tracer Gas Experiments

Calibrating the rooms by releasing CO_2_ yielded unexpected results. Initially, we anticipated the small room would behave as a well-mixed space, with a relatively uniform CO_2_ concentration throughout. However, a sharp decline in CO_2_ concentration in the first two meters from the CO_2_ generator was detected (Figure 6, sensors M1 and M2). Reviewing the sensor data for the small room (Figure 7A1–A3), this is due to low CO_2_ concentration in the distant sensors M3 and M4 in the first run (A1), as if fresh air were displacing CO_2_-laden air, or the current that drives CO_2_ toward the sensors diminished. This pattern does not repeat in the two subsequent runs, A2 and A3, which more closely resemble a mixed model, as all four sensors (M1 to M4) report substantially similar CO_2_ concentrations. 

In the medium room, a tremendous spike (three samples occurring over 3 min; 1164, 2722, 1608 ppm, above baseline) in CO_2_ concentration is observed in the sensor nearest the CO_2_ generator (M1) in run 4 (Figure 7B4). The plume from the generator is very concentrated, so an air current that brought it near the sensor could explain the spike. A similar spike occurred in the Cubicles M1 sensor (Figure 7D1), peaking 900 ppm above baseline.

Another possibility could be the protocol. The experimenter pours the crushed dry ice into the beaker, leaves the room, and records the start of the release when the hair dryer is remotely plugged in. During this loading time, the dry ice is in contact with the room-temperature glass beaker and ambient air, causing some CO_2_ gas to be released and accumulate around the generator. The sudden start of the hair dryer may cause currents that drive the fumes towards the M1 sensors. 

In the large room (Figure 7C1–C4), we detected unexpected peaks at sensor M3 of a magnitude greater than sensor M2, which is closer to the source. We suspect this might be due to turbulent flow caused by the meeting of air from source vents on both sides of the room (see vent locations on Figure 6).

One way to address these fluctuations is to conduct more runs per room. One advantage of using a tank of CO_2_ is that a series of runs can be automated, which is not possible with the dry ice method, as it requires manual loading for each run.

The cubicle mass transfer fraction falls in the middle range of the six rooms studied (Figure 6). Working solo at a desk in a private office for many hours presents low risk, as there is no shared air with others. However, caution is advised when a desk is in a cubicle, especially when only a few agents occupy the space. With less body heat generated, the air exchange rate decreases, leading to poor ventilation. Since agents spend most of their time in these cubicles, seating them close together could result in prolonged exposure to shared air, increasing the risk of airborne transmission. To mitigate this risk, agents should be spread out whenever possible.

We have discussed using CO_2_ sensors for the specific task of room characterization. Once this task is complete, the sensors remain valuable for continuously monitoring risk by measuring CO_2_ levels in occupied meeting spaces [10].

### 4.2. Accounting for More Variables

Room airflow in most commercial buildings is controlled by variable air volume (VAV) systems, which adjust dampers to regulate airflow based on room temperature. Since no people were present to generate heat during tracer gas measurements, the VAV systems operated at their minimum airflow setpoint, resulting in the lowest air exchange rates. This created a worst-case scenario with the highest mass transfer fractions and predicted exposures. By lowering room thermostats to force the VAV dampers open, air exchange rates could increase, allowing for more comprehensive exposure predictions across a range of conditions.

Testing without occupants eliminates variations in CO_2_ readings caused by human presence, such as multiple CO_2_ sources and disruptions in airflow from body heat, physical obstructions, breathing, movement, and door openings [34]. However, to achieve a more realistic and accurate characterization of aerosol dispersion, these variables must be considered. Significant fluctuations in CO_2_ measurements have been observed in several rooms (Medium, Large, and Cubicles) even without humans present. An interesting experiment could involve placing minimally animated mannequins (e.g., rocking back and forth) to simulate human presence.

### 4.3. CO_2_ Generator Evaluation

Both the CO_2_ generator and warm human breath create a vertical rise velocity [34]. However, the higher velocity of the generator may result in underreporting of CO_2_ dispersion in rooms with tall ceilings, such as cafeterias and auditoriums, as the gas is propelled upward and mixes with fresh air that remains undetected by sensors positioned lower in the space.

Dry ice offers several advantages over pressurized CO_2_ containers. It’s widely available, lightweight, easy to transport, and requires no valves or piping, making it safer and simpler to handle. The CO_2_ released can be easily measured by weight. However, gas cylinders allow for automating multiple runs.

To gain greater control over the CO_2_ gas generation rate, velocity, and temperature, an improved generator could use one electric heater to sublimate the dry ice, another to heat the released gas, and a variable-speed fan to regulate flow. Replaceable venting ports could be used to simulate breathing in one, multiple, or all directions.

### 4.4. Browser Evaluation

The interactive browser-based tool offers an efficient, user-friendly way to create and compare policies. Using “Business as Usual” as a baseline provides relatable context while avoiding misleading claims of absolute infection prediction. By modeling human interactions through parameters such as attendance, duration, frequency, and room type, it simplifies use and avoids privacy concerns or the complexity of syncing with calendars and room reservation systems, making it easily adaptable for both individuals and managers.

Fast response to changes in NMI parameters is essential for enabling interactive policy creation. The generic Meeting Profile, based on key factors such as the number of attendees, room size, duration, and intervals, allows for flexible deployment across various work environments. These parameters are common to meetings in diverse settings, including offices, schools, hospitals, construction sites, manufacturing plants, and government agencies. 

The Meeting Profile can be applied to individuals, teams, or groups, each with their own Meeting and Room Profiles to assess how policies impact exposure. Alternatively, a single Meeting and Room Profile could be used across a company to develop example policies, educating employees on relative risks based on meeting types, room size, duration, and attendee numbers. 

### 4.5. Deployment Options

We have described a system using minimal technology to estimate exposure risk. Given the many variables influencing exposure, some of which were encountered during our experiments, a more accurate assessment would require specialist expertise. Small to mid-size air balancing companies are commonly hired by commercial buildings and institutions to optimize HVAC performance, reduce energy costs, and maintain healthy indoor environments. 

Considering how unprepared society was for the COVID-19 pandemic—and possibly still is for future pandemics—these companies could offer a service providing site-specific “pandemic preparedness tools”, similar to the web application described. Accuracy would increase significantly with more runs, generators, and sensors integrated into a sophisticated computational fluid dynamics (CFD) model [35]. The final product could be a user-friendly web application, akin to the one already developed. To generalize across different ventilation systems and environments, consultants would either need to specialize by industry (e.g., hospitals vs. offices) or be large enough to cover a broad range of skills and expertise. Since building ventilation codes vary by country, state, and municipality, consultants would typically focus on specific geographic regions.

### 4.6. Planning for Future Pandemics

Frutos et al. [36] call for a shift in how society approaches emerging infectious diseases. They view the cause as biological—a pathogen—and the spread as societal. Therefore, they argue that it should be addressed not just as a medical problem requiring a medical solution. Instead, it should also be tackled by regulating human activities such as meetings, gatherings, and events that contribute to the spread of the disease. Having site-specific, empirically grounded policies in place before a pandemic provides a mechanism to dynamically modify human behavior to manage the emergence of the disease.

To prepare for future pandemics, a few NMI profiles could be proactively crafted using the interactive tool we described, calibrated to and triggered by color code warnings from local jurisdictions based on case numbers [37]. For example, a four-color alert system—blue, green, yellow, and red—corresponds to ascending infection case numbers and would be paired with corresponding policies. A “blue” status would institute the blue policy with “business-as-usual” (BAU) NMI settings. At the other extreme, “red” status enforces a red policy, requiring mandatory mask wearing and 2-m social distancing. 

## 5. Conclusions

We designed an affordable system for individuals with basic mechanical skills to perform site-specific empirical room airborne mass transfer calibration. We successfully created and demonstrated an end-to-end solution, from site-specific calibration to an interactive application that any user can operate to customize and evaluate the relative impact of non-medical interventions on airborne disease exposure. The simulation highlighted the risks of low-occupancy cubicle spaces, especially when combined with close proximity and extended durations, and demonstrated the benefits of moving meetings to larger rooms and comparing various masking and social distancing strategies.

While we successfully developed a low-cost system to empirically characterize aerosol propagation, we uncovered several significant shortcomings. Specifically, the subsampling of CO_2_ concentration with only four sensors per room and one CO_2_ source and the limited number of runs per room led to overgeneralization and failed to capture location-specific variations in airborne exposure due to the complex, dynamic nature of airflow. We also did not capture increased air exchange rates and dynamic conditions arising from humans in rooms.

Based on our experiences exploring non-medical intervention (NMI) policies with our tool, we gathered the following key insights:Hold meetings in the largest room possible.Maintain two-meter social distancing.Meet and eat outdoors whenever feasible.Monitor airflow in cubicle rooms closely and increase it as much as possible, especially since agents spend a lot of time in their cubicles.Leave the first row empty during presentations, as extended loud talking produces above-average aerosol emissions.Allow rooms to “rest” after use, particularly after long or crowded meetings.Use accurate CO_2_ sensors to spot-check room air quality.When in doubt, wear a mask.

## Figures and Tables

**Figure 1 ijerph-21-01413-f001:**
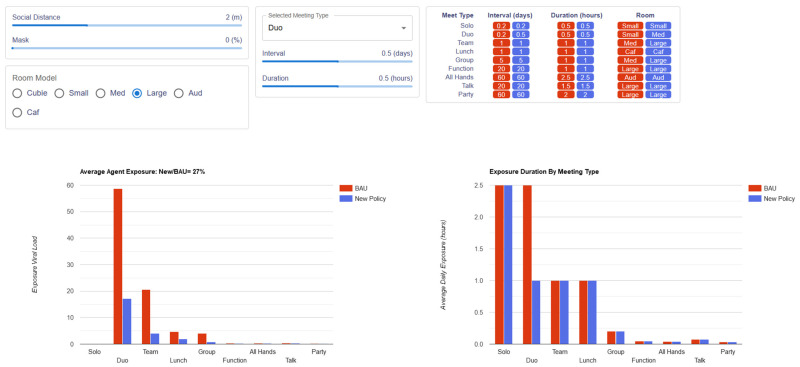
User interface for exposure simulation. Fixed “Business As Usual” (BAU) policy parameters are highlighted in red, while user-adjustable parameters for the new policy are shown in blue. (**Top left**) Two sliders control social distancing (1 to 4 m) and the mask-wearing population (0 to 100%). Below these sliders, six radio buttons allow for the selection of the meeting room. (**Top middle**). A dropdown menu is used to select the meeting type, which adjusts the meeting room interval, duration, and room type. (**Top right**) The interface displays the meeting type, duration, and room settings for both the BAU and user-defined new policies. (**Bottom left**) A bar graph shows the average agent exposure, in arbitrary units of viral load, for each meeting type under the BAU policy (in red) and the user-defined policy (in blue), with the ratio of new to BAU exposure displayed as a performance metric. (**Bottom right**) Bar graph displaying exposure duration, in hours, by meeting type. In this example, agents under the new policy experience 27% of the exposure seen under the BAU policy (a 73% reduction), achieved by upgrading all meetings to larger rooms (where possible) and reducing the intervals of duo meetings from five times a day (interval = 0.2) to twice a day (interval = 0.5).

**Figure 2 ijerph-21-01413-f002:**
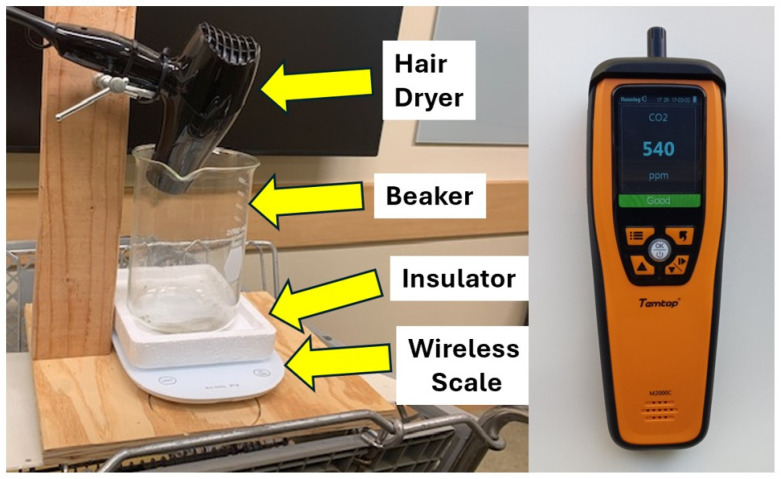
CO_2_ generator and sensors. (**Left**) Components are identified by yellow arrows. CO_2_ generator uses crushed dry ice loaded into a 2 L Pyrex glass beaker. Mass is remotely monitored with wireless scale. A 1500 W hair dryer directs hot air down to sublimate dry ice and mix in hot air. (**Right**) Datalogging CO_2_ sensors (Model M2000C-2nd Generation, Elitech) samples CO_2_ (ppm) once per minute. Data is downloaded over USB interface.

**Figure 3 ijerph-21-01413-f003:**
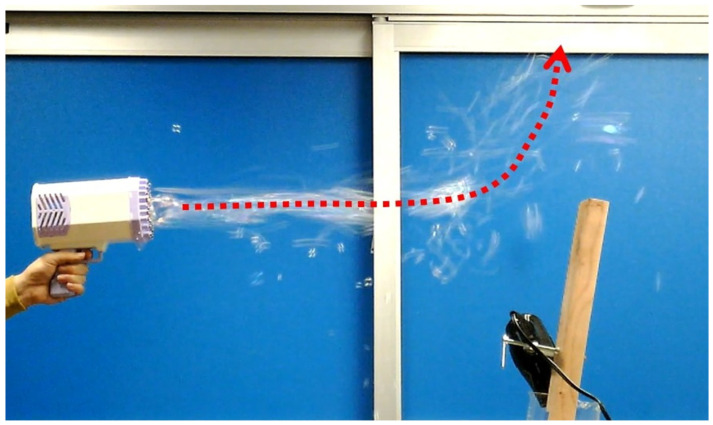
Air flow visualization. The vertical plume from the CO_2_ generator is visualized with soap bubbles. The hair dryer (black object), clamped to a wooden rail, blows hot air into a beaker of crushed dry ice. The dashed red arrow indicates the direction of the resulting airflow.

**Figure 4 ijerph-21-01413-f004:**
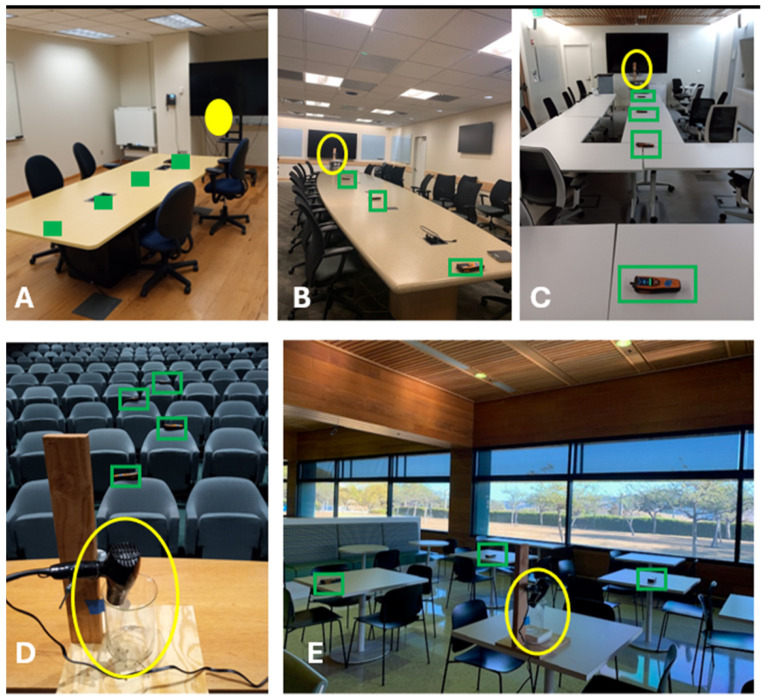
Rooms used to characterize mass transfer fraction using CO_2_ tracer gas. CO_2_ generator is indicated by yellow ovals. CO_2_ sensors are indicated with green rectangles. (**A**) Small room (**B**) Medium room (**C**) Large room (**D**) Auditorium with CO_2_ generator on stage with CO_2_ sensors placed on the top of chairs in successive rows. (**E**) Cafeteria with CO_2_ generator on table, CO_2_ sensors on adjacent tables.

**Figure 5 ijerph-21-01413-f005:**
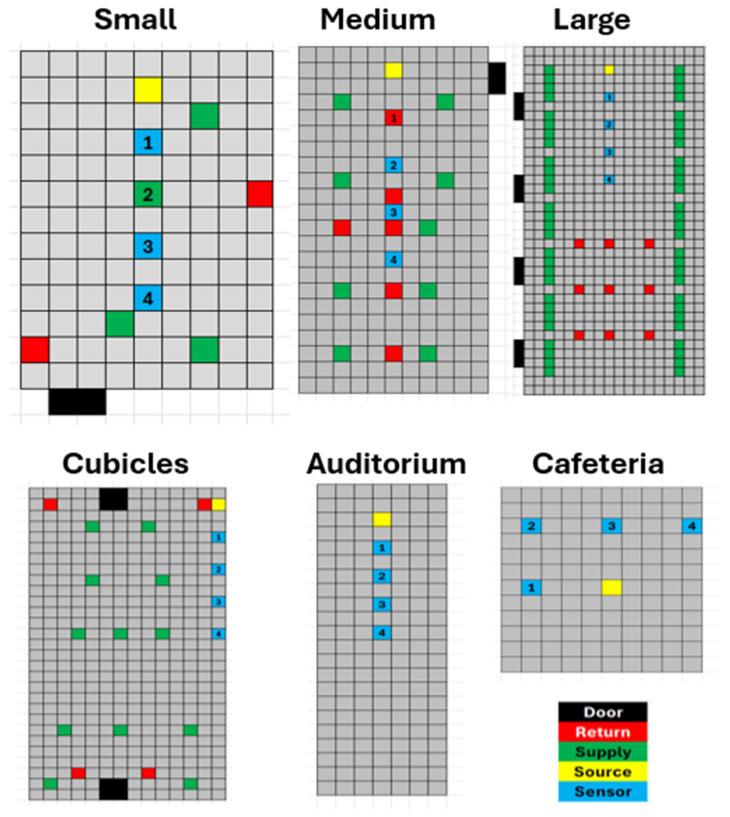
Room floor plans. The rooms are arranged clockwise from the top left: Small, Medium, Large, Cafeteria, Auditorium, Cubicles. The color key is as follows: black represents doors, red indicates the return air (drawn from the room), green indicates the supply air (delivered to the room), yellow indicates the CO_2_ generator, and blue indicates the CO_2_ sensors. The four CO_2_ sensors are indicated by numbers 1, 2, 3, and 4. In the small room (top left), CO_2_ sensor 2 is color-coded green instead of blue, for it is under a supply air duct. Each square represents a 25″ ceiling tile (0.635 m). The locations of all features are rounded to the nearest ceiling tile.

**Figure 6 ijerph-21-01413-f006:**
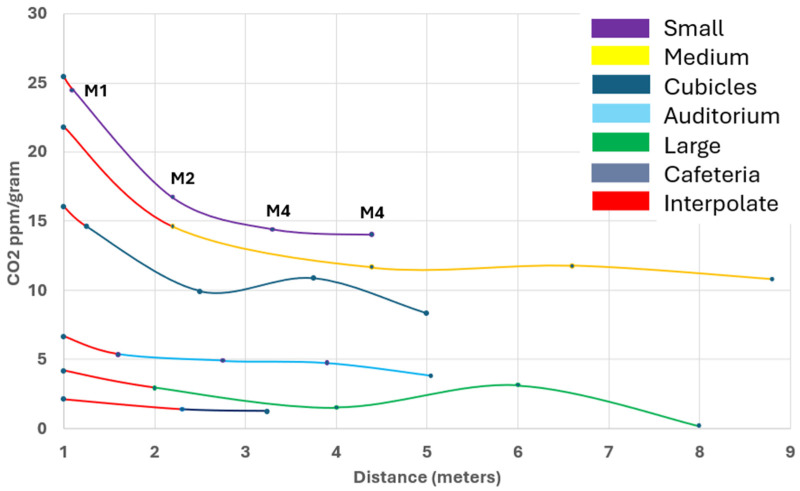
Room mass transfer fractions. Average of accumulated room CO_2_ concentrations (ppm), sampled at locations with CO_2_ sensors, divided by dry ice starting mass. Values extrapolated down to 1 meter (red line segments) using Sun and Zhai [6] infection probability prediction. M1 refers to CO_2_ sensor 1, corresponding to tracer data in Figure 7.

**Figure 7 ijerph-21-01413-f007:**
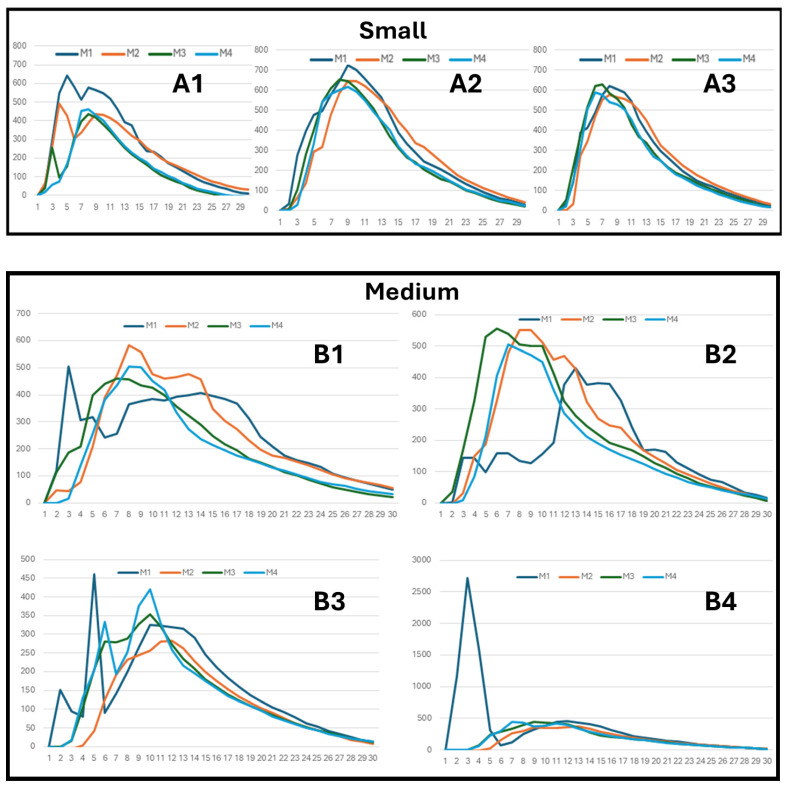
Tracer gas measurements. In each of the six rooms, 400 to 500 grams of dry ice was sublimated with a hair dryer over a period of 10 min. Concentration of CO_2_ gas was measured (ppm) at four locations, nominally 2 meters apart, over a period of 30 min. M1 refers to CO_2_ sensor 1, the closest to the CO_2_ generator and corresponding to M1 in Figure 6. M4 (sensor 4) is the furthest from the CO_2_ generator. The plot notation **A1**, **A2** and **A3** refers to the first, second, and third run in the small room, respectively. This notation is applied to each plot, where the first letter (**A** to **F**) refers to the room, and the number that follows refers to the run.

**Figure 8 ijerph-21-01413-f008:**
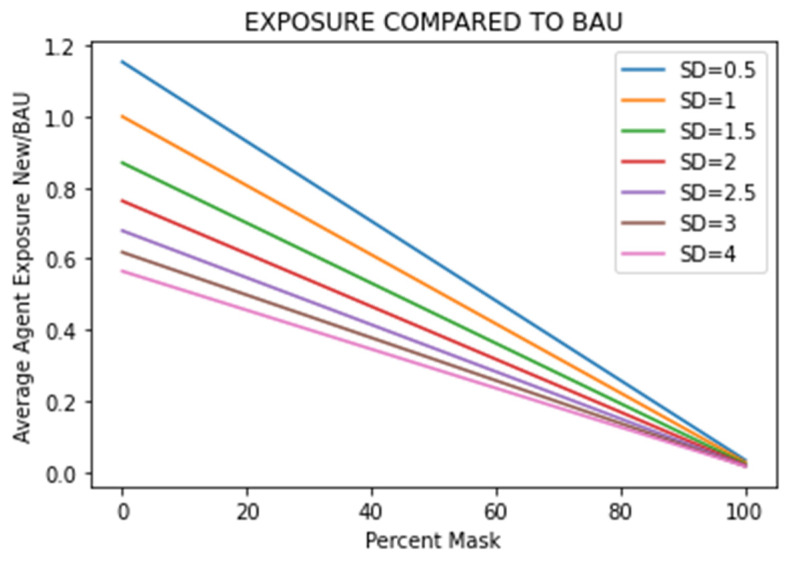
Comparing social distancing and mask-wearing policies.

**Table 1 ijerph-21-01413-t001:** Simulation configuration files. (Left) When deployed at a site, two site-specific files are required. The Meeting Profile represents the typical meeting schedule of an agent. (Right) The Room Profile provides second-degree polynomial coefficients (Ad^2^ + Bd + C, where d is distance) used to calculate the mass transfer fraction, based on CO_2_ release and measurements taken in each room. The seventh room, labeled “Isolated”, indicates that the agent is alone with no exposure to others. MaxD represents the maximum distance for which the coefficients are valid. Beyond this distance, the MaxD value is applied, assuming a mixed-model approach.

Meeting Type	Interval (Days)	Duration (h)		Room	A	B	E	MaxD
Solo	0.2	0.5		Small	1.57	−11.74	35.51	4.5
Duo	0.2	0.5		Med	0.31	−4.21	24.37	8.5
Team	1.5	2		Large	−0.01	−0.33	4.03	8
Lunch	1	1		Auditorium	0.09	−1.11	7.37	5
Group	5	1		Cafeteria	0.19	−1.20	3.14	3.25
Function	20	1		Cubie	0.47	−4.52	19.65	5
All Hands	60	2.5		Isolated	0	0	0	0
Talk	20	1.5						
Party	60	2						
Manager	10	1						

**Table 2 ijerph-21-01413-t002:** Room ventilation specifications. Air changes per hour (ACH) are based on the maximum design flow rate. Notes: * Minimum airflow is 25%, except for the large room, the minimum of which is 50% due to its proximity to a commercial kitchen. ** Average ceiling height.

Room	Area (sq. Feet)	Ceiling Height (Feet)	Max Airflow (cfm)	Min Airflow (cfm)	Room Volume (cf)	Max ACH	Min ACH *
Small	450	9	800	200	4050	12	3
Medium	930	9	850	213	8370	6	1.5
Large	1130	9	1910	955	10,170	11.3	5.6
Cubicles	1580	9	5445	1361	14,220	23	5.8
Auditorium	10,710	13.4 **	10,710	2678	63,930	10	2.5
Cafeteria	6470	16.8 **	12,000	3000	108,372	6.6	1.7

**Table 3 ijerph-21-01413-t003:** The impact of room assignments on relative exposure is examined with the exposure tool. Moving to a larger room decreases exposure. Policy 2–6 conditions: SD = 2, Mask = 0.

Policy	Solo	Duo	Team	Group	New/BAU	Comment
1	Small	Small	Medium	Medium	1	BAU
2	Small	Medium	Medium	Medium	0.82	BAU, SD = 2
3	Small	Small	Medium	Large	0.79	Upgrade Group
4	Small	Medium	Medium	Medium	0.74	Upgrade Duo
5	Small	Small	Large	Medium	0.56	Upgrade Team
6	Large	Medium	Large	Large	0.19	All in Large

## Data Availability

The CO_2_ room data is available on https://github.com/tzim999/CO2_COVID_SIM (accessed on 16 October 2024).

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
