# Peer review of "Interactive Simulation of Nonpharmaceutical Interventions of Airborne Disease Transmission in Office Settings"

_ijerph, 2024, doi:10.3390/ijerph21111413_

Round 1
Reviewer 1 Report
Comments and Suggestions for Authors
Thank you very much for your efforts. I read this article with interest. The article looks quite impressive. However, I have some concerns.
1. Limited Data: Although the simulation is based on CO2 measurements, the claim that CO2 is an accurate representation of virus particles is controversial. Human breath and aerosol distribution are not considered in sufficient detail in this model.
Study 2 produced results for a specific office environment, but its generalizability to different ventilation systems or different environments in different geographic regions is limited. In real world offices, there are many more variables.
3. Social and Psychological Factors Were Ignored: Although the study was technically successful, it does not take into account important factors such as the psychological and social interactions of employees. There is no discussion of how measures such as social distancing and mask-wearing affect employee performance.
4. The experimental setup used in the study (e.g., use of the COâ‚‚ generator and measurement setup) is based on a specific office environment. There is no sufficient explanation for the applicability of the same model to different office environments, ventilation systems or geographical conditions. It should be noted how adaptations can be made in different environments.
5. Realism of Working Conditions: Conducting experiments in an environment where people are not present (for example, in an environment with minimal ventilation) may not accurately reflect real-world office conditions. Factors such as human movements, opening of doors and the dynamic operation of ventilation in real office environments are ignored.
6. Analysis of the effectiveness of mask use and social distancing was limited to only a few distances (1, 2, 3 meters) and mask ratios. Different scenarios (for example, greater distances, higher mask use rates, or changing airflow) could have been considered in more detail. Additionally, more detailed analyzes such as differences between mask types (e.g., surgical mask vs. N95) were not conducted.
7. The findings obtained in the Discussion section had to be considered in a broader context to provide a solution to the general problem stated at the beginning of the study (the spread of airborne diseases in office environments). More comprehensive comments would have been expected, for example, on how to prepare for future pandemics or in which other sectors the results could be applied.
Author Response
Thank you very much for your efforts. I read this article with interest. The article looks quite impressive. However, I have some concerns.
- Limited Data: Although the simulation is based on CO2 measurements, the claim that CO2 is an accurate representation of virus particles is controversial. Human breath and aerosol distribution are not considered in sufficient detail in this model.
AUTHORS COMMENTS: We added “Section 1.1 CO2 as a surrogate for aerosolized virus” to discuss these issues, along with several references. Ai e.tl. raised some good objection and addresses them. In the Discussion we acknowledge that the absence of humans limits the realism of our approach and make some suggestions going forward in the new “Section 4.2 Accounting for more variable” we added.
- Study 2 produced results for a specific office environment, but its generalizability to different ventilation systems or different environments in different geographic regions is limited. In real world offices, there are many more variables.
AUTHORS COMMENTS: Throughout the document we now emphasize that the tracer gas calibration is done at every new site, so it is customized to a site. In response to your concern, we added “Section 4.5 Deployment options” to address this issue. It also motivated us to add section 4.6 “Planning for future pandemics”.
- Social and Psychological Factors Were Ignored: Although the study was technically successful, it does not take into account important factors such as the psychological and social interactions of employees. There is no discussion of how measures such as social distancing and mask-wearing affect employee performance.
AUTHORS COMMENTS: That's an important observation. Great point. We added Section 1.3 “Social impacts of NMI”.
- The experimental setup used in the study (e.g., use of the COâ‚‚ generator and measurement setup) is based on a specific office environment. There is no sufficient explanation for the applicability of the same model to different office environments, ventilation systems or geographical conditions. It should be noted how adaptations can be made in different environments.
AUTHORS COMMENTS: We now address those issues in new sections 4.4 Browser evaluations and 4.5 Deployment options.
- Realism of Working Conditions: Conducting experiments in an environment where people are not present (for example, in an environment with minimal ventilation) may not accurately reflect real-world office conditions. Factors such as human movements, opening of doors and the dynamic operation of ventilation in real office environments are ignored.
AUTHORS COMMENTS: Yes, we acknowledge this in new section 4.2 Accounting for more variables and 4.5 Deployment options and the limitations of our approach for the reasons you pointed out in Conclusions.
- Analysis of the effectiveness of mask use and social distancing was limited to only a few distances (1, 2, 3 meters) and mask ratios. Different scenarios (for example, greater distances, higher mask use rates, or changing airflow) could have been considered in more detail. Additionally, more detailed analyzes such as differences between mask types (e.g., surgical mask vs. N95) were not conducted.
AUTHORS COMMENTS: We added 4 more distances to Figure 8, extending to 0.5 meters and 4 meters. We added Section 1.2 discussing surgical and N95 masks, and the pointed out the big problem is the fit of the mask. We ran the simulation with N95 but it didn’t make much difference using lab tests that say surgery masks are 85% and 95% effective on infection (receiving) and transmission, respectively. What’s more important, as we now point out in the paper, is people wearing a mask (surgery or N95) and making sure it fits well.
- The findings obtained in the Discussion section had to be considered in a broader context to provide a solution to the general problem stated at the beginning of the study (the spread of airborne diseases in office environments). More comprehensive comments would have been expected, for example, on how to prepare for future pandemics or in which other sectors the results could be applied.
AUTHORS COMMENTS: Exactly. Thank you for waking me up to this point. I found a great paper by Frutos #37 in my references discussed in the new Section 4.6 “Planning for future pandemics”. I also added Section 4.5 Deployment options.
Reviewer 2 Report
Comments and Suggestions for Authors
1. The author needs to reference a larger number of sources in their citations.
2. Section 1 needs to include more literature and undergo significant revisions.
3. Add a summary of the overall contributions of the paper at the end of Section 1.
4. It is recommended to divide Figure 5 into four separate images or reorganize it, as it appears to be missing a 5-minute interval.
5. In the conclusion, it is suggested to add a section on future research directions.
Comments on the Quality of English LanguageModerate editing of English language required.
Author Response
- The author needs to reference a larger number of sources in their citations.
AUTHORS COMMENTS: We agree! More than doubled the number of citations, from 17 to 38. Much more background on the complexity of the problem, and where to go from here.
- Section 1 needs to include more literature and undergo significant revisions.
AUTHORS COMMENTS: We agree. Three new sections added 1.1, 1.2, and 1.3.
- Add a summary of the overall contributions of the paper at the end of Section 1.
AUTHORS COMMENTS: The last paragraph of the introduction now lists four contributions of the paper.
- It is recommended to divide Figure 5 into four separate images or reorganize it, as it appears to be missing a 5-minute interval.
AUTHORS COMMENTS: There are now 21 separate images because I plotted all the data and refer to them in a new Discussion section 4.1 Analyzing tracer gas experiments.
- In the conclusion, it is suggested to add a section on future research directions.
AUTHORS COMMENTS: Excellent idea. See new sections 4.3 CO2 generator evaluations for ideas on a new generator, 4.5 Deployment options and 4.5 Planning for future pandemics!
Reviewer 3 Report
Comments and Suggestions for Authors
Interactive simulation of airborne disease transmission
The manuscript presents a simple experimental procedure to estimate aerosol transmission between occupants of different size rooms.
As the authors mentioned in the discussion, the air change rate in a room has important influence on the mass transfer of aerosols. At a minimum, they should report the air change rate for their experiments. They should also report the type of ventilation system and the approximate geometry and location of diffusers and return ducts. In future studies, the use of smoke pencils to determine approximate air circulation patterns would be helpful.
The experiments were carried out in empty rooms. Several studies have shown that thermal plumes set up by heat exchange from individuals can have a first-order influence on the overall air circulation pattern. This includes the rise velocity of aerosol-laden plumes from infected individuals. Visual estimates of CO2 plume vertical velocities should be compared to results in the literature for typical thermal plume velocities from individuals.
The approximate nature of this experimental technique should be mentioned in their discussion.
Comments on the Quality of English Language
no comments
Author Response
The manuscript presents a simple experimental procedure to estimate aerosol transmission between occupants of different size rooms.
As the authors mentioned in the discussion, the air change rate in a room has important influence on the mass transfer of aerosols. At a minimum, they should report the air change rate for their experiments. They should also report the type of ventilation system and the approximate geometry and location of diffusers and return ducts. In future studies, the use of smoke pencils to determine approximate air circulation patterns would be helpful.
AUTHORS COMMENTS:
(a) Air change rate: We now report the air change rate.
(b) Type of ventilation system and diffuser/return duct locations: We added floor plans indicating where the inlet and outlets are.
(c) Smoke pencils to visualize air circulation: We included visualizing the air flow of the CO2 generator with tiny soap bubbles, new Figure 3.
The approximate nature of this experimental technique should be mentioned in their discussion.
AUTHORS COMMENTS: We added Section 4.2 Accounting for more variables to address some of the limitations.
Round 2
Reviewer 2 Report
Comments and Suggestions for Authors
Accept in present form
Reviewer 3 Report
Comments and Suggestions for Authors
follow up work with suggestions in new section